# Thermal impacts of engineering activities and vegetation layer on permafrost in different alpine ecosystems of Qinghai-Tibet Plateau, China

Wu Qingbai[1, 2] Zhang Zhongqiong[1] Gao Siru[1] Ma Wei[1,2]

[1]State Key Laboratory of Frozen Soil Engineering, Cold and Arid Regions Environmental and Engineering Research Institute, Chinese Academy of Science, Lanzhou, 730000, China
[2]Beiluhe Observation Station of Frozen Soil Environment and Engineering, Cold and Arid Regions Environmental and Engineering Research Institute, Chinese Academy of Science, Lanzhou, 730000, China

*Correspondence to*: Wu Qingbai (qbwu@lzb.ac.cn)

**Abstract.** Climate warming and engineering activities have various impacts on the thermal regime of permafrost in alpine ecosystems of the Qinghai–Tibet Plateau. Using recent observations of permafrost thermal regimes along the Qinghai–Tibet Highway and Railway, the changes of such regimes beneath embankments constructed in alpine meadows and steppes are studied. The results show that alpine meadows on the Qinghai–Tibet Plateau can have a controlling role

among engineering construction effects on permafrost beneath embankments. As before railway construction, the artificial permafrost table (APT) beneath embankments is not only affected by climate change and engineering activities but is also controlled by alpine ecosystems. However, the change rate of APT is not depending on ecosystems type, which are predominantly affected by climate change and engineering activities. Instead, the rate is mainly related to cooling effects of railway ballast and heat absorption effects of asphalt pavement. No large difference between alpine

and steppe can be identified regarding the variation of soil temperature beneath embankments, but this difference is readily identified in the variation of mean annual soil temperature with depth. The vegetation layer in alpine meadows has an insulation role among engineering activity effects on permafrost beneath embankments, but this insulation gradually disappears because the layer decays and compresses over time. On the whole, this layer is advantageous for alleviating permafrost temperature rise in the short term, but its effect gradually weakens in the long term.

**1 Introduction**

Climate warming and engineering activities significantly impact permafrost thermal regimes on the Qinghai–Tibet Plateau (Cheng and Wu, 2007; Jin et al., 2008; Wu and Zhang, 2008; Zhang et al., 2008; Yang et al., 2010). However, the response of permafrost to climate warming differs greatly from that of engineering construction (Wu et al., 2007). This difference is mainly caused by permafrost thermal stability (Wu et al., 2007).

Because permafrost is the result of energy and mass exchange between the ground surface and atmosphere, its response to climate warming and engineering activities is modulated by ground surface conditions, e.g., vegetation, soil, and geological conditions (Brown et al., 2000; Hinkel and Nelson, 2003; Frauenfeld et al., 2004). Permafrost has a close relationship with alpine ecosystems, and changes of permafrost can significantly affect those ecosystems (Callaghan and Jonasson, 1995; Jorgenson et al., 2001; Hinzman et al., 2005; Wang et al., 2006; Shur and Jonasson, 2007; Gregory et al., 2012; Wang et al., 2012). Climate warming has varying thermal impacts on permafrost in different alpine ecosystems (Wu et al., 2015), change in permafrost temperature and active layer thickness (ALT) of alpine meadows is greater than that of alpine steppe (Wu et al., 2015). Therefore it is a concern, whether engineering activities have thermal impacts on permafrost that vary with ecosystems. Further, removing or retaining vegetation in highway or railway construction may cause differences in permafrost change and engineering stability. However, there has been little research in this area.

Engineering activities on the Qinghai–Tibet Plateau, e.g., the Qinghai–Tibet Highway (QTH) and Railway (QTR), resulted in a substantial increase of permafrost temperatures, rise of the permafrost table, and thawing of ground ice near the permafrost table beneath embankments (Wu et al., 2002; Sheng et al., 2002; Ma et al., 2009; Wu et al., 2010b; Mu et al., 2012). However, the cited research works treated only the thermal disturbance impacts of the highway or railway on permafrost beneath embankments. There has been little attention to interaction among engineering activities, vegetation or soils near the ground surface, and permafrost beneath embankments, an exception being McHattie and Esch (1983) who studied the permafrost benefits of a peat underlay in roadway construction.

The main objective of the present study was to investigate the thermal impacts of engineering activities on permafrost beneath embankments in various ecosystems, using data and information from a continuous record of permafrost temperature monitoring along the QTH and QTR corridor. We first focus on the analysis of annual means and variability of the artificial permafrost table (APT) beneath embankments in alpine meadows and steppes over the period 1996/2005 through 2014. We then investigate trends of soil temperature and various impacts of ecosystems in driving changes of soil temperature. Finally, we assess the advantages and disadvantages of removing vegetation during engineering construction.

## 2 Data and method

The soil temperature data used were obtained from nine monitoring sites along the QTH and QTR (Fig. 1). Because vegetation was removed when the QTH was constructed but remained present when the QTR was constructed, data were obtained from six centerline boreholes beneath the highway embankment (three in alpine steppe and three in alpine meadow) and six beneath the railway embankment (four centerline boreholes in alpine meadow and two shoulder boreholes in alpine steppe). For comparison, soil temperature from a borehole beneath a natural surface was obtained from the same location as the centerline borehole beneath the embankment for all sites.

**2.1 Site description**

Soil temperature was measured at 12 sites from the Chumaer high plain in the north to the Tanggula Mountains in the south, with six sites along the QTH and six sites along the QTR (Fig. 1). Five boreholes were drilled in the Chumaer high plain (two along the railway and three along the highway), four in the Beiluhe Basin (railway), and three in the Fenghuo, Kaixinling, and Tanggula Mountains (highway). Geographic information and ecosystems of these sites are listed in Tables 1 and 2.

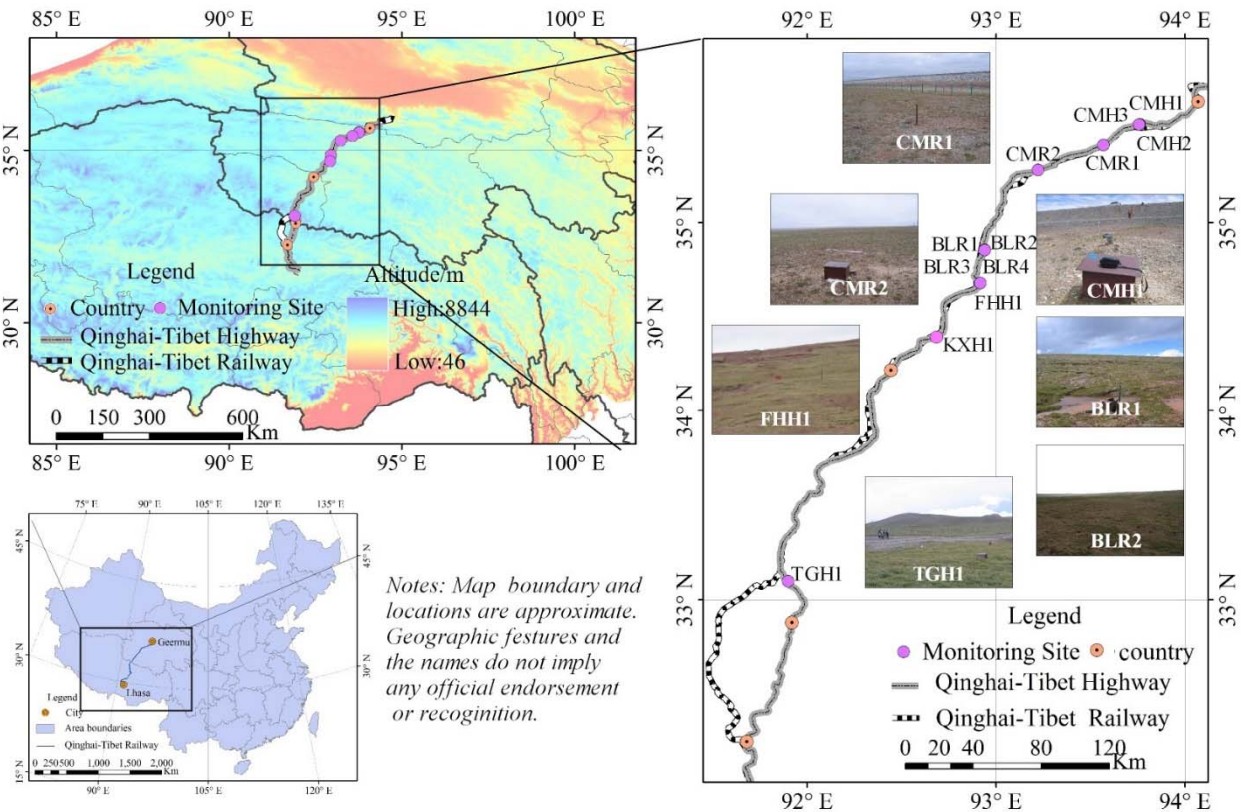

**Figure 1 Geographic locations of 12 monitoring sites**

Mean annual air temperature ranges from about −5.0 to −5.5 °C in the Chumaer high plain to about −6.0 to −6.5 °C in the Tanggula Mountains (Table 2). The climate of the Qinghai–Tibet Plateau is extremely continental with annual total precipitation generally < 300 mm, but this precipitation varies from 230 to 510 mm in the Beiluhe Basin (Wu et al., 2015). There was no steady or winter-long snow cover at any site during the study period.

Alpine grassland along the QTH and QTR mainly includes alpine meadow and alpine steppe (Wang et al., 2006). Five monitoring sites in the Chumaer high plain are in alpine steppe, with 5–10 % vegetation cover at CMH1, CMH2 and CHM3, and 20–30% at CMR1 and CMR2 (Fig. 1). Four sites in the Beiluhe Basin are in alpine meadow, with vegetation cover 60–70% at BLR1, 30–40% at BLR2, 60–70% at BLR3, and 80–90% at BLR4. Another three monitoring sites, in the Fenghuo,

Kaixinling, and Tanggula Mountains, respectively, are also in alpine meadow, with vegetation cover ~85–90% at FHH1, 35–40% at KXH1 and ~93–97% at TGH1. Near-surface sediments are dominated by gravel and sandy soil in alpine steppe, and clayey and silt soil with little gravel in alpine meadow. Soil organic content is relatively low where there is no peat layer in alpine steppe, and high where there is such a layer (< 10 cm).

**Table 1 Information on 12 monitoring sites along QTH and QTR**

| NO. | Site* | Areas | Latitude (°) | Longitude (°) | Altitude (m) | Observation Period |
|-----|-------|-------|--------------|---------------|--------------|--------------------|
| 1 | CMH1 | | 35.52 | 93.76 | 4577 | 2002–2014 |
| 2 | CMH2 | Chumaer High Plain | 35.52 | 93.76 | 4572 | 2002–2014 |
| 3 | CMH3 | | 35.52 | 93.76 | 4568 | 2002–2014 |
| 4 | CMR1 | | 35.41 | 93.57 | 4477 | 2005–2014 |
| 5 | CMR2 | | 35.28 | 93.22 | 4583 | 2005–2014 |
| 6 | BLR1 | | 34.86 | 92.92 | 4633 | 2002–2014 |
| 7 | BLR2 | Beiluhe Basin | 34.85 | 92.94 | 4632 | 2002–2014 |
| 8 | BLR3 | | 34.85 | 92.94 | 4630 | 2002-2014 |
| 9 | BLR4 | | 34.82 | 92.92 | 4654 | 2002-2014 |
| 10 | FHH1 | Fenghuo Mts | 34.68 | 92.92 | 4950 | 1996–2014 |
| 11 | KXH1 | Kaixinling | 33.96 | 92.35 | 4627 | 2003-2014 |
| 12 | TGH1 | Tanggula Mts. | 33.10 | 91.90 | 4948 | 2002–2014 |

*XXR indicates sites from QTR and XXH sites from QTH

**2.2 Soil temperature measurements**

All monitoring sites were established in 2002, except for FHH1 in 1996 and CMR1 and CMR2 in 2005. Soil temperature measurements from all sites are continuous through the present. Soil temperature was measured at depths 0.5–18 m beneath the surface of the embankment centerline at all sites, except for that beneath the sunny-side shoulder at CMR1 and CMR2. All measurements were made by a string of thermistors at depth increments of 0.5 m. These thermistors were made by the State Key Laboratory of Frozen Soil Engineering. Laboratory temperature accuracy of these sensors is ±0.05 °C. For all sites along the QTH, in-situ measurements of soil temperatures were conducted by well-trained technicians using data loggers (CR3000, Campbell Scientific Inc., USA), with automatic measurements on the 5th and 20th days of each month. For all sites along the QTR, in-situ measurement data were automatically collected each day by the data logger at 10:00 a.m. Beijing Standard Time.

**2.3 Method**

We analyzed the long-term trends and variability of APT, ALT, and permafrost temperature beneath embankments in alpine meadow and steppe from 2002 through 2014. APT (ALT) was estimated as the maximum thaw depth in late autumn, using linear interpolation of soil temperature profiles between two neighboring points near and below the 0 °C isotherm beneath

embankments at all sites. Long-term trends of APT were estimated by linear regression, using 13 years of APT data at each site, with $p < 0.01$. Long-term trends of annual mean permafrost temperature were estimated by half-monthly measurements for QTH and daily measurements for QTR. Long-term variability of permafrost temperature was estimated via linear regression, using 13 years of annual mean permafrost data at each site, with $p < 0.01$.

**Table 2 Climate and environmental parameters at 12 monitoring sites along QTH and QTR**

| No. | Site | Climate Conditions * | | Ecosystem | VC ( %) | Permafrost Conditions | | |
| | | MAAT (°C) | Precipitation (mm) | | | MAGT (°C) | ALT (m) | FST |
|---|---|---|---|---|---|---|---|---|
| 1 | CMH1 | | | | 5–10 | | | B,H |
| 2 | CMH2 | | | | 5–10 | −1.45 | 1.79 | B,H |
| 3 | CMH3 | −5.0 to −5.5 | 230–250 | Alpine Steppe | 5–10 | | | B,H |
| 4 | CMR1 | | | | 20–30 | −0.91 | | B,H |
| 5 | CMR2 | | | | 20–30 | −0.22 | 4.88 | D,F |
| 6 | BLR1 | | | | 60–70 | −0.97 | 1.76 | H |
| 7 | BLR2 | −3.4 to −4.0 | 230–510 | Alpine Meadow | 35–40 | −0.57 | 2.23 | H |
| 8 | BLR3 | | | | 60-70 | -1.07 | 1.96 | H |
| 9 | BLR4 | | | | 80–90 | -1.04 | 1.97 | H |
| 10 | FHH1 | −6.0 to −6.5 | 250–300 | Alpine Meadow | 85–90 | −2.13 | 2.02 | F, B |
| 11 | KXH1 | −3.4 to −3.8 | 250–480 | | 35–40 | −0.92 | 1.64 | F, B |
| 12 | TGH1 | −6.0 to −6.5 | 250–300 | | 93–97 | −1.14 | 2.02 | F, B |

MAAT: mean annual air temperature; ALT: active layer thickness; MAGT: mean annual ground temperature at depth of zero annual amplitude, usually at 10–15 m depth below ground surface on the plateau; FST: frozen soil types, where H stands for frozen soils with ice, B for saturated frozen soils, F for saturated frozen soils with excess ground ice, and D for icy soils; VC: vegetation cover.

*Climate conditions from Zhao et al., 2004; Wu et al., 2012; Wu et al., 2015.

## 3 Results

### 3.1 Change in APT beneath embankments

Mean APT beneath embankments in alpine steppe between 2002–2014 ranged from 6.5 m at CMR1 and CMH2 to 7.83 m at CMH3, with an average of 7.03 m (Table 3). In comparison, mean APT beneath embankments in alpine meadow between 2002 (1995 at FHH1) and 2014 ranged from 3.39 m at BLR3 to 8.43 m at KXH1, with an average of 4.86 m, resulting in a difference of mean APT between alpine meadow and steppe of 2.2 m. This difference is similar to result below natural surfaces in alpine meadow and alpine steppe (Wu et al., 2010c; Zhao et al., 2010; Li et al., 2012). For alpine meadow, mean APT beneath QTH embankments (6.47 m) was larger than that beneath QTR embankments (3.65 m). However, for alpine steppe,

APT beneath QTH embankment (average 7.22 m) was only slightly larger than that beneath QTR embankment (average 6.74 m).

ALT beneath natural surfaces increased continuously between 2000–2014 along the QTH and QTR (Fig. 2) due to climate warming (Wu and Zhang, 2010; Wu et al., 2012; Zhao et al., 2010). The annual ALT rate of increase ranged from 1.79 cm/a at CMR2 to 5.45 cm/a at FHH1, with an average of 3.54 cm/a (Table 3). As seen for the annual ALT rate of increase in the Beiluhe Basin (Wu et al., 2015), the rate in alpine meadow, which varied from 3.53 cm/a at TGH1 to 5.45 cm/a at FHH1 with an average of 4.29 cm/a, was greater than that in alpine steppe. The latter ranged from 1.79 cm/a at CMR2 to 2.31 cm/a at CMH2, with an average of 2.05 cm/a (Table 3), yielding a difference of mean ALT rate of increase between alpine meadow and alpine steppe of more than 2.0 cm/a.

**Table 3 Rate of change for artificial permafrost table (APT)**

| Site Name | APT change beneath Embankment | | | | ALT change beneath nature surface | | | |
|---|---|---|---|---|---|---|---|---|
| | Max (m) | Min (m) | Mean (m) | Rate (cm/a) | Max (m) | Min (m) | Mean (m) | Rate (cm/a) |
| Ecosystem | Alpine steppe | | | | | | | |
| CMH1 | 8.06 | 6.40 | 7.33 | 11.17 | | | | |
| CMH2 | 7.06 | 5.52 | 6.50 | 8.14 | 1.94 | 1.62 | 1.79 | 2.31 |
| CMH3 | 8.39 | 6.16 | 7.83 | 14.26 | | | | |
| CMR1 | 7.53 | 4.90 | 6.50 | −31.78 | | | | |
| CMR2 | 7.70 | 6.58 | 6.98 | −7.45 | 4.98 | 4.75 | 4.88 | 1.79 |
| Ecosystem | Alpine meadow | | | | | | | |
| FHH1 | 3.84 | 3.19 | 3.56 | −2.06 | 2.35 | 1.2 | 1.75 | 5.45 |
| TGH1 | 8.40 | 6.08 | 7.43 | 19.20 | 2.26 | 1.82 | 2.02 | 3.53 |
| KXH1* | 8.74 | 7.81 | 8.43 | 10.30 | 1.89 | 1.42 | 1.64 | 4.35 |
| BLR1 | 4.96 | 3.53 | 4.09 | −9.95 | 2.1 | 1.44 | 1.76 | 3.68 |
| BLR2 | 4.41 | 3.36 | 3.62 | −7.12 | 2.49 | 1.95 | 2.23 | 4.46 |
| BLR3 | 4.2 | 3.32 | 3.50 | −5.13 | | | | |
| BLR4 | 5.83 | 2.46 | 3.39 | −28.9 | | | | |

KXH1*, trend analysis range from 2004 to 2011 because thermosyphon was installed in 2010.

Figure 3 shows that there is a substantial difference between the QTH and QTR in annual APT change rate, and that this difference is independent of alpine meadow or alpine steppe. For the QTH, APT beneath embankments predominantly increased except at FHH1 (where it decreased at an annual rate of −2.26 cm/a) (Fig. 3a). This rate of increase varied from 8.14 cm/a at CMH2 to 19.20 cm/a at TGH1, with an average of 12.6 cm/a. For the QTR, APT beneath embankments continuously decreased (Fig. 3). The annual rate of decrease ranged from −31.78 cm/a at CMR1 to −7.12 cm/a at BLR2, with an average of −15.1 cm/a. This great difference in annual APT change rate between the QTH and QTR is attributed to strong heat absorption by the asphalt pavement (Sheng et al., 2002; Wu et al., 2010a), cooling of railway ballast pavement (Lai et al., 2003; Cheng et al., 2007; Ma et al., 2008), and strengthening measures of the QTR after 2007 (Hou et al., 2015). As a consequence, no clear effect of the influence of alpine meadow or alpine steppe on interannual APT change can be seen.

**3.2 Changes in permafrost temperature**

Because vegetation was removed during QTH construction but present during QTR construction, it is important to analyze the effect of the vegetation layer on the soil thermal regime in alpine meadow and steppe for both constructions. Therefore, we examined the long-term variability of soil temperature at 0.5 m beneath the embankment base, near-permafrost table temperature, and permafrost temperature at a depth of 10 m (see Fig. 4).

Figure 5 shows changes of mean annual soil temperature at 0.5 m depth beneath the embankment base along the QTH and QTR in alpine meadow and steppe. Mean annual soil temperature at that depth had a decreasing trend along the QTR, but show an increasing trend along the QTH except for a decrease at FHH1. At QTR sites, this rate of change varied from −0.16 °C/decade at BLR2 to −1.5 °C/decade at CMR1 with an average of −0.48 °C/decade (Table 4). At QTH sites except FHH1, the rate of change ranged from 0.76 °C/decade at CMH3 to 1.24 °C/decade at TGH1, with an average of 0.92 °C/decade (Table 4).

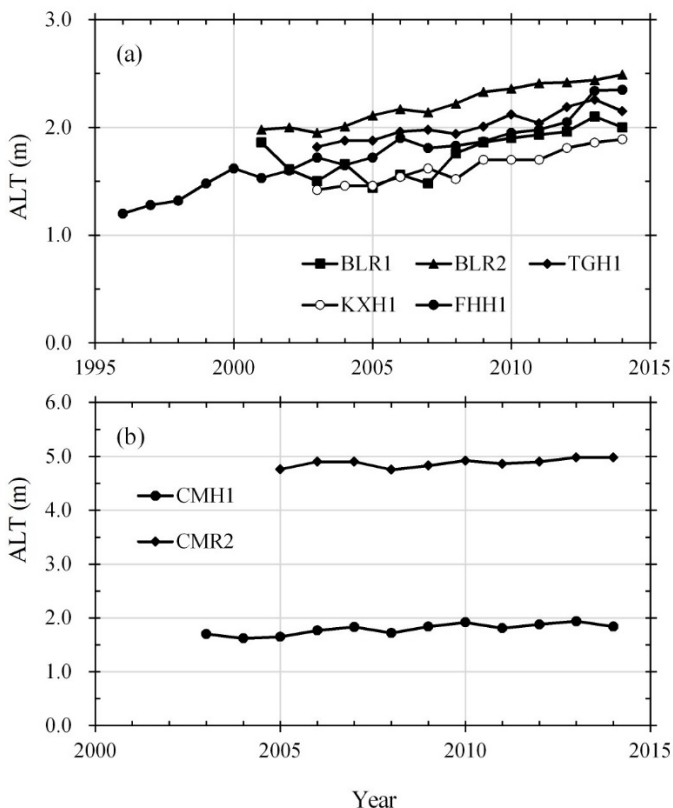

**Figure 2 Active layer thickness (ALT) beneath natural surface in alpine meadow (a) and alpine steppe (b)**

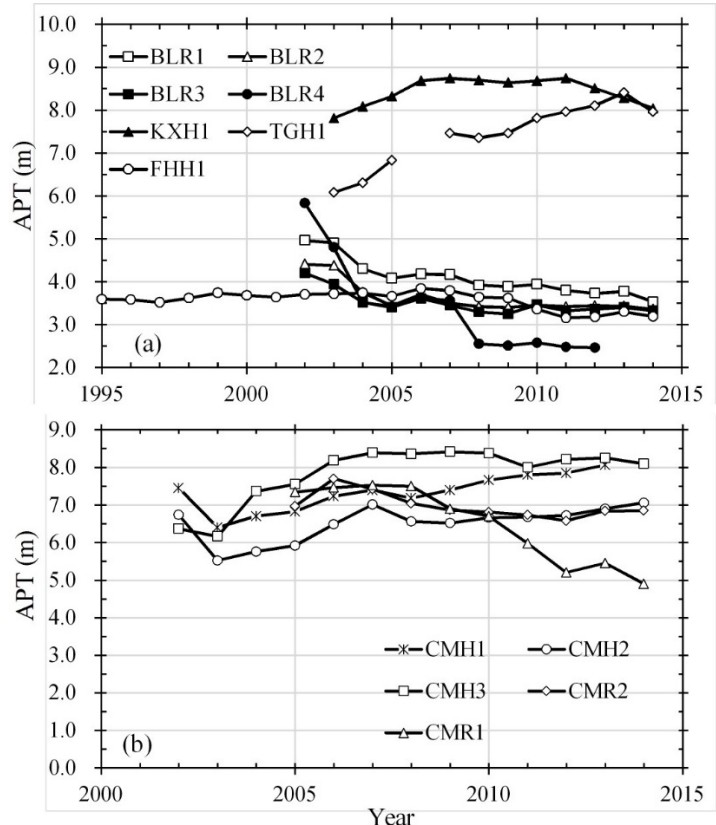

**Figure 3 Artificial permafrost table (APT) beneath embankment in alpine meadow (a) and alpine steppe (b)**

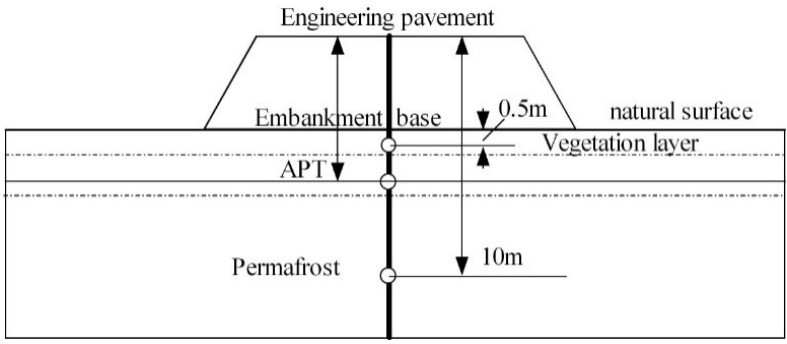

**Figure 4 Schema of soil temperature measurements at (i) 0.5 m depth beneath the embankment, (ii) near the artificial permafrost table, and (iii) at 10 m depth.**

Figure 6 shows changes of near-permafrost table temperature beneath embankments along the QTH and QTR in alpine meadow and steppe. Over the observation period, this temperature showed a decreasing trend along the QTR, but an increasing trend along the QTH except for a decrease at FHH1. At QTR sites, the rate of near-permafrost table temperature decrease varied from −0.01 °C/decade at CMR2 to −0.39 °C/decade at BLR1 and BLR3, with an average of −0.21 °C/decade (Table 4).

At QTH sites except FHH1, the rate of increase ranged from 0.13 °C/decade at CMH3 to 0.38 °C/decade at TGH1, with an average of 0.27 °C/decade.

**Table 4 Variation rate of mean annual soil temperature at 0.5 m depth beneath embankment base (EBT0.5), near-permafrost table temperature (NPT), and permafrost temperature at depth 10 m (PT10) beneath embankment**

| Site Name | Change rate of soil temperature beneath embankment, °C/10a | | | Observation Period |
|---|---|---|---|---|
| | EBT0.5 | NPT | PT10 | |
| Ecosystem | Alpine Steppe | | | |
| CMH1 | 0.91 | 0.31 | 0.41 | 2002–2014 |
| CMH2 | 0.77 | 0.23 | 0.17 | 2002–2014 |
| CMH3 | 0.76 | 0.13 | 0.09 | 2002–2014 |
| CMR1 | −1.50 | −0.19 | - | 2005–2014 |
| CMR2 | −0.23 | −0.01 | 0.07 | 2005–2014 |
| Ecosystem | Alpine Meadow | | | |
| FHH1 | −0.42 | −0.22 | 0.19 | 2002–2014 |
| TGH1 | 1.24 | 0.38 | 0.19 | 2002–2014 |
| KXH1* | - | 0.32 | 0.26 | 2004-2014 |
| BLR1 | −0.45 | −0.39 | 0.44 | 1996–2014 |
| BLR2 | −0.16 | −0.11 | 0.17 | 2002–2014 |
| BLR3 | −0.19 | −0.18 | 0.39 | 2003-2014 |
| BLR4 | −0.35 | −0.39 | 0.33 | 2003-2011 |

KXH1*: trend analysis range from 2004 to 2011, because thermosyphon was installed at the end of 2010.

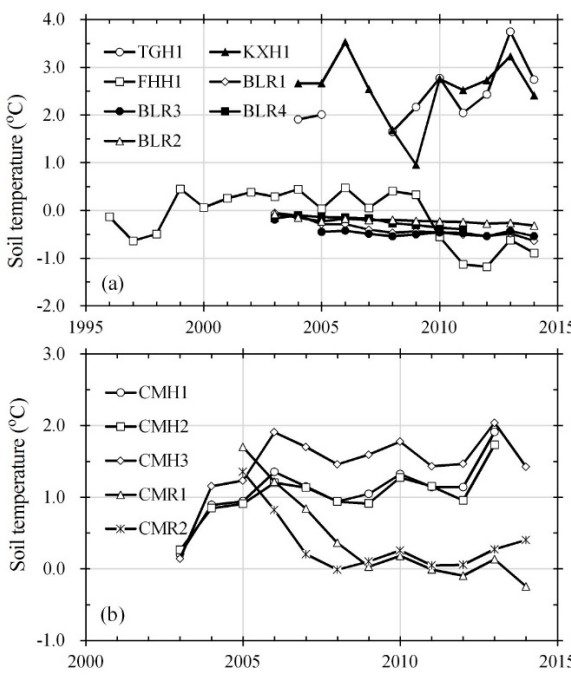

**Figure 5 Mean annual soil temperature at 0.5 m depth beneath embankment base in alpine meadow (a) and alpine steppe (b)**

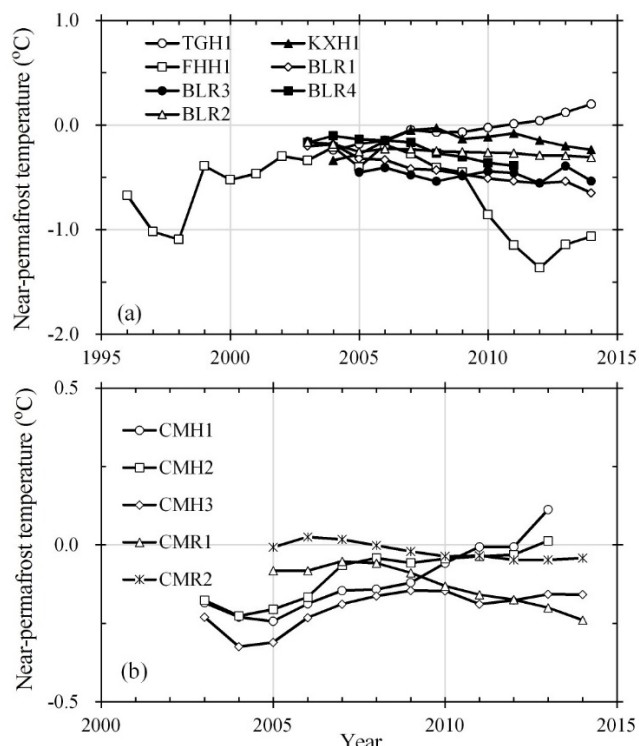

**Figure 6 Near-permafrost table temperature (cf. Figure 4) beneath embankment in alpine meadow (a) and alpine steppe (b)**

Figure 7 shows changes of permafrost temperature at 10 m depth beneath the embankment surface of the QTH and QTR,
5  in alpine meadow and steppe. Contrary to the shallow temperatures shown in Figure 5 and 6, the 10 m temperature shows an
increasing trend at almost all sites. At the QTR sites, the rate of permafrost temperature increase at 10 m depth varied from
0.07 °C/decade at CMR2 to 0.44 °C/decade at BLR1, with an average of 0.28 °C/decade; however, there was no obvious
increasing trend at CMR1 (Table 4). At the QTH sites, the rate of temperature increase ranged from 0.09 °C/decade at CMH3
to 0.41 °C/decade at CMH1, with an average of 0.22 °C/decade.
10  The results shown in Figure 5−7 showed that changes of soil temperature at 0.5 m depth beneath the embankment base
and near-permafrost table temperatures could not clearly be related to alpine meadow or alpine steppe, but rather to air
temperature increase on the one hand, and the cooling of QTR ballast pavement and heat absorption of QTH asphalt pavement
on the other hand. Changes of permafrost temperature at 10 m depth were independent of engineering type (QTH or QTR),
and their rate of increase approached 0.28 °C/decade for QTR and 0.22 °C/decade for QTH. Notably, the mean temperature
15  increase at this depth was higher in alpine meadow (0.25 °C/decade) than in alpine steppe (0.18 °C/decade), and had a similar
pattern to that of temperature beneath a natural surface (Wu et al., 2015). The thermal effect of engineering activities on
permafrost gradually weakened with time.

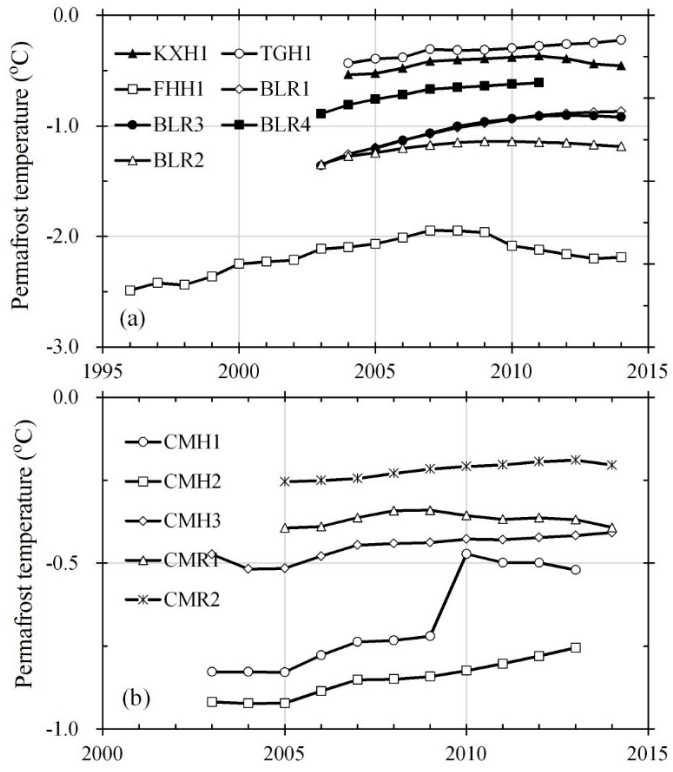

**Figure 7 Permafrost temperature at 10 m depth beneath embankment surface in alpine meadow (a) and alpine steppe (b)**

## 4. Discussion

Vegetation in alpine ecosystems is important for freezing-thawing processes and the permafrost thermal regime (Zhang
5  et al., 2005; Shur and Jorgenson, 2007; Hinzman et al., 2005; Wang et al., 2012). The response of permafrost in different alpine
ecosystems to climate change varies greatly (Wu et al., 2015). Because the vegetation layer beneath embankments can be
important for the change of APT and the permafrost thermal regime, the effect of engineering activities on permafrost likely
varies with the alpine ecosystem. However, we cannot infer that changes of APT, daily mean soil temperature, and permafrost
temperature are closely related to the influence of the vegetation layer in alpine meadow and steppe based on our study.
10  Therefore, we analyzed variations of soil temperature with depth beneath embankments in alpine meadow and steppe. Figure
8 shows variations of mean annual soil temperature with depth beneath the embankments at BLR1 (a) and BLR2 (b) for the
QTR, where a vegetation layer in alpine meadow was present, and at TGH1 (e), FHH1 (f) and KXH1 (g) for the QTH, where
this layer was removed. From this figure it can be seen that for BLR (with vegetation layer) the mean annual soil temperature
gradually approached an almost isothermal value with depth (about −0.77 °C at BLR1 in (a) and −0.36 °C at BLR2 in (b)).
15  However, there was no such finding (Figure 8e, f and g) for the QTH without a vegetation layer. Under the combined effect
of climate change and engineering activities, soil temperature in the upper vegetation layer shows a distinct decreasing trend,

but whereas temperature at a certain depth range beneath the vegetation layer had an increasing trend for the railway with a vegetation layer in alpine meadow (Figures 8a and b and Table 4). In contrast, soil temperature at all depths showed increasing trends for the highway with removed vegetation layer (Figure 8e, f and g and Table 4). These differences indicate that the vegetation layer in an alpine meadow may have an insulating role among the effects of engineering activities on permafrost beneath embankments. Similarly, we analyzed changes of the permafrost thermal regime beneath embankments in alpine steppe, for both QTR with a vegetation layer and QTH without that layer. The results showed that no pattern similar to that in Figure 8a and b was present, demonstrating that that vegetation layer has no insulation effect in this case. We conclude that the vegetation layer in an alpine meadow can effectively prevent heat disturbance from engineering construction from propagating rapidly downward and raising permafrost temperature over the short term, which is not the case for a vegetation layer in alpine steppe. This is because a vegetation layer in alpine meadow contains humus soils with small thermal conductivity, reducing downward heat propagation.

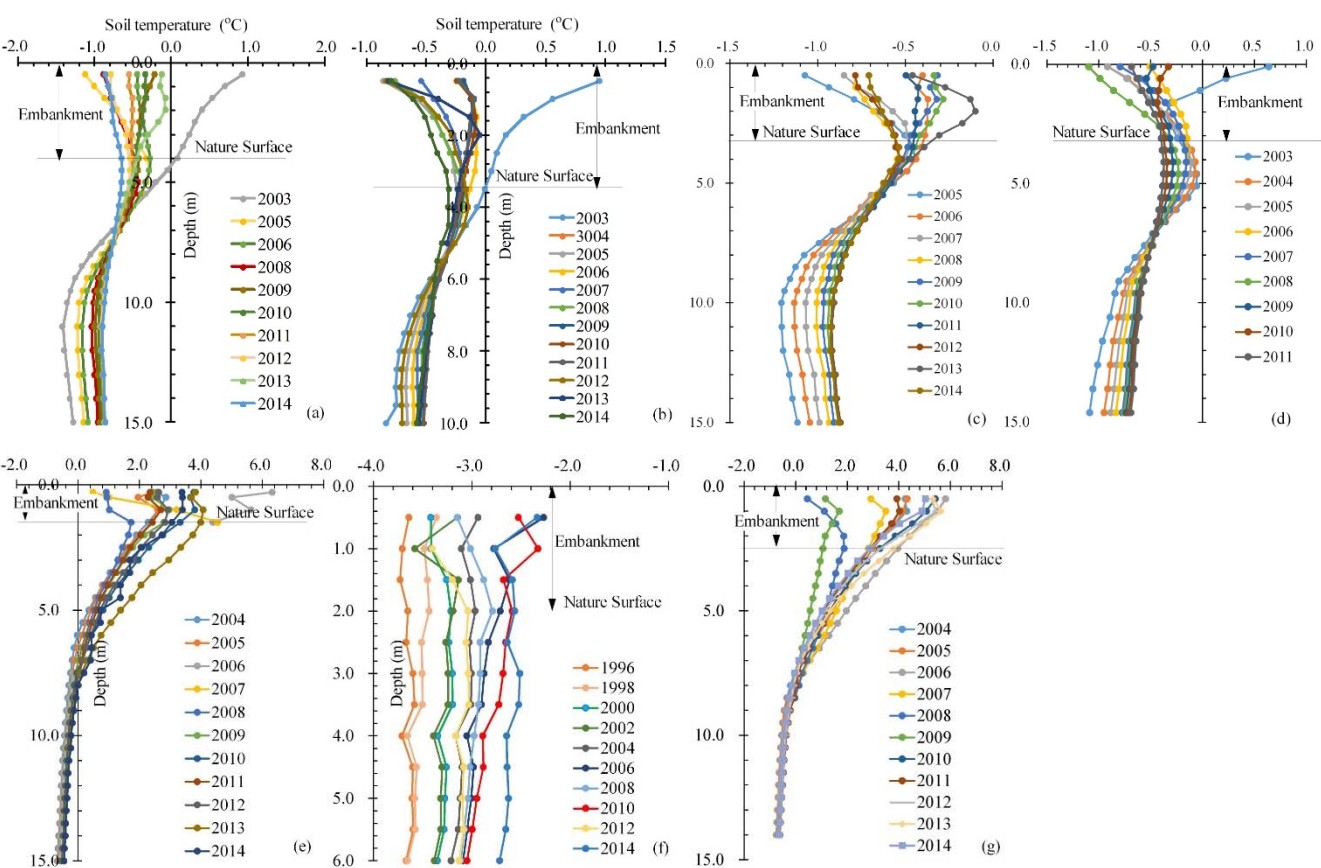

**Figure 8 Change of mean annual soil temperature with depth beneath the embankments at BLR1 (a) and BLR2 (b), BLR3 (c), and BLR4 (d) for QTR, and at TGH1 (e), FHH1 (f), and KXH1 (g) for QTH in alpine meadow**

Generally, an insulation layer within an embankment can mitigate heat disturbance from short-term engineering activities (Esch, 1987; Cheng et al., 2004; Wen et al., 2005). However, such a layer is a disadvantage regarding long-term effects of climate warming over the period of engineering operation (Liu et al., 2002; Sheng et al., 2006), especially for warm permafrost because an insulation layer cannot prevent warm permafrost from thawing within long-term. Although we cannot know what happens to an insulating vegetation layer after it is buried under a railroad grade, we can infer that this layer is compressed over time, altering its thermal properties in an alpine meadow. As a consequence, its insulation effect may gradually weaken (McHattie and Esch, 1983). From Fig. 8 it can be seen that the temperature gradient from the vegetation layer to a given depth beneath the embankment gradually decreases with time, and the trend of permafrost warming gradually weakens between 3 and 4 years after railway construction. This indicates that the heat insulation effect of vegetation changes.

Generally, the vegetation layer in alpine meadow of the Qinghai-Tibet Plateau, including the humus and root-layer soils, is thin, with maximum thickness < 60 cm (Li et al., 2007). Vegetation roots mainly reach depths of ~10 cm, and mean root biomass makes up 60% of total root biomass (Yue et al., 2015). After the railway or highway embankment is constructed, soil within the vegetation layer of the alpine meadow is compressed and soil moisture decreases, modifying soil heat transfer. Because heat conductivity within the vegetation layer of alpine meadow from humus soil is less than that of filled soil of embankments, the vegetation layer can effectively prevent downward heat transfer, decreasing the amount of heat in permafrost. Meanwhile, moisture within the alpine meadow vegetation layer migrates upward into the embankment soil. At present, we cannot quantitatively analyze such a process, and more research is needed to quantify its effect on the thermal regime. In addition, heat and moisture transport through the vegetation layer may be affected by lateral heat transfer, different geometries of the roadbed/railroad, and the presence of snow on the lateral embankment slopes.

The effect of lateral heat transfer on permafrost beneath embankments can have two sources: horizontal conductive heat exchange between embankment and outside air, and convective processes within the embankment slope. The horizontal heat exchange is generally small due to low horizontal heat conduction. However, heat transfer by lateral convection strongly influences permafrost beneath embankments. Water flow can especially accelerate permafrost thaw (Grandpré et al., 2012). The heat effect of embankment slopes on permafrost beneath the embankment is mainly due to the differential input of solar radiation on the sunny and shaded slopes of the embankment (Chou et al., 2008a). The resulting difference in solar radiation produces differences in soil temperature and the permafrost table under the shoulder (Chou et al., 2008b; Wu et al., 2011). Monitoring data of soil temperature along the QTR show that the differences in temperature and APT between sunny and shaded slopes of the embankment at WD3, KL1 and KL3 in alpine meadow (Wu et al., 2012) are generally small, < 1 °C and 20 cm, respectively, Wu et al., 2011), but that difference in alpine steppe is > 1.5–3.0 °C and 100–300 cm, respectively (Wu et al., 2011). These results may indicate that the alpine meadow vegetation layer beneath embankments reduces differences in soil temperature and APT under the shoulder.

Varying geometries of the roadbed/railroad have a thermal effect on permafrost beneath the embankment. For example, a large embankment height will increase that difference, because of larger radiation input on the sunny slope (Hu, 2006). The embankment width affects the annual heat transfer rate at the bottom of the embankment (Yu et al., 2007). The annual rate

increased by 60% with doubling of the width of asphalt pavement (Yu et al., 2007). This increased rate was mainly at the bottom of the embankment, resulting in a concentration of heat, which then enters the permafrost though the vegetation layer.

On the Qinghai-Tibet Plateau, snow mainly accumulates in the high mountains, with only little snow in the plateau interior (Li and Mi, 1983; Sun et al., 2014). Snow cover along QTH and QTR is therefore generally thin, with less than 6 cm on

average, and a short snow cover duration of cover (Li and Mi, 1983; French, 2007; Tian et al., 2014). The insulation of snow cover is weak when it is < 20 cm in thickness (Zhang, 2005; Jin et al., 2008). Although there is no steady snow cover in winter on the plateau, snow accumulation at the foot of embankment slopes is possible, with thickness < 20 cm. Thus, snow accumulation at the foot of the slope may have no large effect on permafrost beneath the embankment. However, thaw of accumulated snow increases soil moisture at the foot of the slope.

In general, ground temperatures in permafrost regions of the Qinghai-Tibet Plateau are mainly controlled by regional climate conditions, as indicated by strong regional zonation of elevation, latitude, and continentality (Cheng, 1982). The temperatures are also greatly affected by local factors such as vegetation, snow cover, sand cover and surface conditions. These influences can increase or decrease ground temperature under certain circumstances (Jin et al., 2008).    The regional and local factors can cause significant offsets between mean annual air temperature (MAAT) and mean annual ground temperature

(MAGT) (Zhou et al., 2000; Wang, et al., 2002). However, engineering surfaces such as asphalt pavement cause anomalously high surface temperatures through radiative heating. This causes a difference between MAAT and mean annual ground surface temperature, generating accelerated permafrost degradation under the embankment (Wu et al., 2011; Zhang et al., 2016).

## 5 Conclusions

Based on soil temperature observations at nine monitoring sites over the period 2002/2004 through 2014 along the QTH

and QTR, we studied the variation of APT and soil temperature beneath embankments. The results show that alpine ecosystems on the Qinghai–Tibet Plateau can alter the effects of engineering construction on permafrost beneath embankments. Average APT beneath embankments was between 4.68 m at alpine meadow sites and 7.03 m at alpine steppe sites. However, the variation rate of APT was not closely related to the alpine ecosystem but rather to the engineering type (railroad/highway). APT beneath QTH embankments showed an increasing trend, with an average of 13.2 cm/a. In contrast, APT beneath QTR

embankments showed a decreasing trend, with an average of −14.1 cm/a. Our findings indicate that alpine ecosystems can affect APT magnitude beneath embankments but do not influence the rate of APT change. That rate is more related to the cooling of railway ballast and heat absorption of asphalt pavement.

Soil temperature at a depth of 0.5 m and near-permafrost table temperature beneath embankments in QTH alpine ecosystems showed an increasing trend over the observation period, with averages of 0.84 °C/decade and 0.26 °C/decade,

respectively. However, a decreasing trend was found for QTR, with averages of −0.60 °C/decade and −0.18 °C/decade respectively. Permafrost temperature at a depth of 10 m beneath the embankment surface had an increasing trend, with an average of 0.22 °C/decade. The changes in soil temperature at a depth of 0.5 m and near-permafrost table temperature are

closely related to the cooling of railway ballast and heat absorption of asphalt pavement, but were unrelated to variation between alpine meadow and steppe. The rate of permafrost temperature increase at a depth of 10 m in alpine meadow (0.25 °C/decade) was slightly higher than that in alpine steppe (0.18 °C/decade).

Changes in mean annual soil temperature with depth beneath embankment surfaces in alpine meadow with a vegetation layer differed from that without a vegetation layer. This suggests that the vegetation layer of alpine meadow has an insulation role within the effects of engineering activities on permafrost beneath embankment, but this insulation gradually disappeared as the vegetation layer decayed and compressed over time. Overall, that layer is an advantage for alleviating permafrost temperature rise in the short term, but its impact gradually weakens in the long term.

## Acknowledgments

This study was supported by the National Natural Science Foundation of China (Grant 41330634), National Key Scientific Research Project (Grant 2011CB026106), and STS Project of the Chinese Academy of Sciences (HHS-TSS-STS-1502).

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
