# Peer review of "Thermal impacts of engineering activities and vegetation layer on permafrost in different alpine ecosystems of Qinghai-Tibet Plateau, China"

_The Cryosphere, 2015_

## Referee Comment (RC1) · Anonymous Referee #1 · 18 Feb 2016

**Review of the Manuscript: Thermal impacts of engineering activities on permafrost in different alpine ecosystems in Qinghai-Tibet Plateau, China by Q. Wu et al.**
**MS No.: tc-2015-218**

**General Comments**

The authors demonstrate the role of vegetation layer in long-term permafrost change beneath embankments, e.g., Qinghai-Tibet Highway and Railway. They analyze the permafrost changes including the artificial permafrost table (APT), permafrost temperatures at APT and the depth of 10 m over a period of 2002/2004-2014, and conclude that the preserved vegetation layer in the alpine meadow will have adverse effect on the permafrost beneath embankments in the long term, although it can alleviate permafrost temperature rise in the short term.

I generally agree with the authors' conclusions, which are based on the valuable long-term in-situ observations. The findings are of great importance for the permafrost engineering community. However, more rigorous arguments should be provided as recommended in the specific comments below. This would further consolidate the paper's conclusions. I therefore recommend the paper for publication, pending the revisions detailed below.

**Specific Comments**

1. The title deviates from the focus of the study. The dominant thermal impact of engineering activities on permafrost origins from the embankment itself (e.g., type), while the vegetation layer should be secondary issue. A more precise title is required to address the role of vegetation layer.

2. Line 157-161 in Section 3.2: an increasing trend of permafrost at a depth of 10 m beneath embankments in both alpine meadow and alpine steppe systems is deduced for overall observations at all sites by using a linear regression. However, concerning the delaying response of permafrost temperature at 10-m depth to previous climate warming and later engineering effects, the regression can mislead the trend. For instance, after an evident increasing trend, a slight decreasing trend occurs after around 2010 at sites FHH1 and BLR2 in Fig. 7a, CMR1 and CMH1 in Fig. 7b. The effect of engineering activities at these sites might be over that of climate. Otherwise, it means the temperature-controlling measures for the QTR failed at these sites. Please clarify the sentence in Line 168-169. This point is a major comment.

3. Line 178-189 in Section 4: the variation of soil temperature with depth beneath embankments in the alpine meadow is related to the isolation effect of the vegetation layer. The thermal isolation effect of the vegetation layer in natural ground usually origins from shielding of radiation and variably thermal properties. However, how

well this mechanisms work beneath the embankment are not introduced in the study, which is essential to the conclusion. This point is a major comment. Please clarify.

4. The terms of vegetation layer and alpine ecosystems are misused in the text, and the later is confusing when used for the layer beneath embankment. Please revise it.

5. Line 130-133: Comparing to the secondary role of vegetation, the difference in embankment type should play a dominant role in influencing soil thermal regime. How do you distinguish the effect of vegetation layer with the primary factor? Add explanation as line 175-188.

**Technical Corrections**

1. Table 1 in P3: add space in "Altitude(m)", and correct the altitude value for CMR2.
2. Table 2 in P4: add sources for the values of climate conditions.
3. Line 76: "Figure 1" --> Fig. 1. Same problems in other places.
4. Line 88 and 90: "in situ" --> in-situ
5. Line 88-91: one datalogger used at all sites? How simultaneously collect at different sites?
6. Line 109: "decrease" --> reduce
7. Line 111: "with average 3.54 cm/s" --> with an average of 3.54 cm/s. Same in other places.
8. Line 110-114: any comments for the different warming rates between alpine meadow and alpine steppe?
9. Line 123-126. "This great difference in annual APT change rate between the QTH and QTR contributed to strong heat absorption by asphalt pavement ..." --> This great difference in annual APT change rate between the QTH and QTR is attributed to strong heat absorption by asphalt pavement ..."
10. Line 126: "Another contribution was engineering activity increase of interannual APT variation beneath embankments" --> not clear.
11. Table 4: "change rate of soil temperature beneath Embankment, °C/10a" --> Change rate of soil temperature beneath embankment, °C/10a
12. Figure 4 in P9: please detail the caption.
13. Line 197: "Based on soil temperature data of nine monitoring sites over the period ..." --> Based on soil temperature observations at nine monitoring sites over the period ...
14. Line 203: "These findings indicate that alpine ecosystems can control APT magnitude beneath embankments but cannot control the rate of APT change" --> the controlling factor on APT magnitude is the alpine ecosystem? Why not climate or embankment?
15. Line 226: "Callaghan, T.V., Jonasson, S.: ..." --> Callaghan, T.V.,  and Jonasson, S.:. Similar error occurs in several references.

16. Line 255: "Li, R., ZHAO L.,..." --> Li, R., Zhao L.,...
17. Please revise carefully the references as required style.

---

## Author Comment (AC1) · 1 Mar 2016

Reply comments on reviewer 1#:

**Specific comments**

1. The title deviates from the focus of the study. The dominant thermal impact of engineering activities on permafrost origins from the embankment itself (e.g., type), while the vegetation layer should be secondary issue. A more precise title is required to address the role of vegetation layer.

Reply: Thank you. I still find a more precise title to reflect the issue of our manuscript. Title is revised as following:
Thermal impacts of engineering activities and vegetation layer on permafrost in different alpine ecosystems in Qinghai-Tibet Plateau, China

2. Line 157-161 in Section 3.2: an increasing trend of permafrost at a depth of 10 m beneath embankments in both alpine meadow and alpine steppe systems is deduced for overall observations at all sites by using a linear regression. However, concerning the delaying response of permafrost temperature at 10-m depth to previous climate warming and later engineering effects, the regression can mislead the trend. For instance, after an evident increasing trend, a slight decreasing trend occurs after around 2010 at sites FHH1 and BLR2 in Fig. 7a, CMR1 and CMH1 in Fig. 7b. The effect of engineering activities at these sites might be over that of climate. Otherwise, it means the temperature-controlling measures for the QTR failed at these sites. Please clarify the sentence in Line 168-169. This point is a major comment.

Reply: Thank you for your opinions. But, these sites which we chose is general embankment without no measures, thermal disturbance of engineering activities is gradually becoming lower. So, the decreasing trend after 2010 is attributed to climate change. So, we revised into:

While, the effect of engineering activities on permafrost is gradually becoming lower. Therefore, the effect of climate warming on permafrost at that depth beneath embankment might be stronger than that of engineering activities.

3. Line 178-189 in Section 4: the variation of soil temperature with depth beneath embankments in the alpine meadow is related to the isolation effect of the vegetation layer. The thermal isolation effect of the vegetation layer in natural ground usually origins from shielding of radiation and variably thermal properties. However, how well this mechanisms work beneath the embankment are not introduced in the study, which is essential to the conclusion. This point is a major comment. Please clarify.

Reply: Thank you. We are neglect to this problem. We add some explanation:

This is because the vegetation layer in an alpine meadow has thicker humus soils with a small thermal conductivity, reducing heat amount conduct down.

4. The terms of vegetation layer and alpine ecosystems are misused in the text, and the later is confusing when used for the layer beneath embankment. Please revise it.

Reply: Thank you. We read in detail the text, we revised misuse terms.

5. Line 130-133: Comparing to the secondary role of vegetation, the difference in embankment type should play a dominant role in influencing soil thermal regime. How do you distinguish the effect of vegetation layer with the primary factor? Add explanation as line 175-188.

Reply: Thank you. We add some explanation as following:
Under overlapping effect of climate change and engineering activities, soil

temperature upper the vegetation layer has an obvious deceasing trend, but soil temperature at the range of definite depth beneath the vegetation layer has an obvious rising trend for railway with the vegetation layer in alpine meadow (Figure 8a, 8b and Table 4). However, soil temperature in all observation depth beneath show obvious rising trend for highway removing vegetation layer (Figure 8c, 8d and Table 4).

Technical Corrections:

1. Table 1 in P3: add space in "Altitude(m)", and correct the altitude value for CMR2.

Reply: We revise into Altitude (m) and correct the altitude of 45.83 to 4583 in Table 1

2. Table 2 in P4: add sources for the values of climate conditions.

Reply: We add two references for the value of climate conditions, Zhao et al., 2004; Wu et al., 2012; Wu et al., 2015.

3. Line 76: "Figure 1" --> Fig. 1. Same problems in other places.
4. Line 88 and 90: "in situ" --> in-situ

Reply: We revised.

5. Line 88-91: one datalogger used at all sites? How simultaneously collect at different sites?

Reply: one or two data loggers used at every sites, for all sites, data are corrected at 10:00 a. m. Beijing Standard Time. We explain this problems in P5, Lines 90-91.

6. Line 109: "decrease" --> reduce
7. Line 111: "with average 3.54 cm/s" --> with an average of 3.54 cm/s. Same in other places. Line 110.

Reply: We revised.

8. Line 110-114: any comments for the different warming rates between alpine meadow and alpine steppe?

Reply: We add a sentence in Line 114.

The difference of mean ALT increasing rate between alpine meadow and alpine steppe is more than 2.0m/a.

9. Line 123-126. "This great difference in annual APT change rate between the QTH and QTR contributed to strong heat absorption by asphalt pavement ..." --> This great difference in annual APT change rate between the QTH and QTR is attributed to strong heat absorption by asphalt pavement ..."

Reply: Thank you. We revised.

10. Line 126: "Another contribution was engineering activity increase of interannual APT variation beneath embankments" --> not clear.

Reply: Thank you, this sentence seems to be repeated. We cancel it.

11. Table 4: "change rate of soil temperature beneath Embankment, °C/10a" --> Change rate of soil temperature beneath embankment, °C/10a.

Reply: We revised.

12. Figure 4 in P9: please detail the caption.

Reply: We revised into:

Figure 4 Soil temperature at 0.5 m depth beneath embankment, near artificial permafrost table and at 10 m depth.

13. Line 197: "Based on soil temperature data of nine monitoring sites over the period ..." --> Based on soil temperature observations at nine monitoring sites over the period ...

Reply: Thank you. We revised.

14. Line 203: "These findings indicate that alpine ecosystems can control APT magnitude beneath embankments but cannot control the rate of APT change" --> the controlling factor on APT magnitude is the alpine ecosystem? Why not climate or embankment?

Reply: Thank you. Except the effect of climate change and embankment on APT, Alpine ecosystem can influence APT magnitude beneath embankment but cannot affect the change rate of APT. So, we revised into:

These findings indicate that alpine ecosystems can influence APT magnitude beneath embankments but cannot affect the change rate of APT, except the effect of climate change and embankment.

15. Line 226: "Callaghan, T.V., Jonasson, S.: ..." --> Callaghan, T.V., and Jonasson, S.:. Similar error occurs in several references.

Reply: Thank you. We revised all references.

16. Line 255: "Li, R., ZHAO L.,..." --> Li, R., Zhao L.,...

Reply: We revised.

17. Please revise carefully the references as required style.

Reply: Thank you. We revised carefully the references as required style.

---

## Referee Comment (RC2) · Anonymous Referee #2 · 3 Mar 2016

The manuscript" Thermal impacts of engineering activities on permafrost in different alpine ecosystems in Qinghai-Tibet Plateau, China" presents measurements of the ground thermal regime in a range of settings impacted by construction activities. Although the manuscript contains a number of interesting data sets and findings, I do not recommend the manuscript for publication in TC, unless serious revisions are conducted by the authors. In particular, the authors present a number of interpretations for their measurements which are not well enough supported. This could be done by a) a statistical analysis of a sufficient number of samples (= boreholes in this case which is most likely difficult to achieve), or b) a careful, at least semi-quantitative argumentation involving the physics of the system, in particular the heat fluxes related to the thermal

properties of the different parts/layers of the system and possibly the radiative forcing at the surface, or c) a study with a 2-dimensional ground thermal model considering the points raised under b).

Major points:

-The manuscript presents and discusses a lot of measurements in great detail (which is OK). However, some of the conclusions on the underlying processes, although not implausible, are not directly supported by the measurements. Although a number of boreholes exist, their number is too small for a statistical analysis that could secure the interpretations given by the authors. On the other hand, no data on the processes themselves are presented. In the following, I comment on the different results and conclusions as given in the abstract (l. 12 ff): "The results show that alpine meadows on the Qinghai–Tibet Plateau can have a controlling role within engineering construction effects on permafrost beneath embankments. " Why not also alpine steppes? Their controlling role would be different, but they would still have a controlling role?

"The artificial permafrost table (APT) beneath embankments is predominantly controlled by alpine ecosystems, . . ." Is this not a direct consequence of different ALTs before construction?

". . . but the change rate of APT is not closely related with those ecosystems. " Is it possible to draw this conclusion from the few boreholes, considering that the spatial variability could be quite high?

"it is mainly related with cooling effects of railway ballast and heat absorption effects of asphalt pavement." No evidence for this is presented, although it is a plausible conclusion.

"Variation of soil temperature beneath embankments is independent of alpine ecosystems, but variation of mean annual soil temperature with depth is closely related to those ecosystems." It is not clear to me what the authors mean with that and how this

could be explained in terms of changing energy content of the soil.

"The vegetation layer in alpine meadows can have an insulation role within engineering activity effects on permafrost beneath embankments. " The problem is that the data set does not allow differentiating between the effects of road pavement/railroad grade vs. vegetation removal/no vegetation removal. It is at least possible that the described effects are entirely due to the different heat transfer processes in the roadbed and railroad grade.

"This insulation role is an advantage for alleviating permafrost temperature rise in the short term," I agree with this finding, the around ten-year time series supports this.

" but a disadvantage in the long term because of climate warming, suggesting that vegetation layer in alpine meadow should be removed upon initiating engineering construction" No evidence for this conclusion is presented.

-What happens to the insulating vegetation layer when it is buried under the railroad grade? How thick is this vegetation layer, and how would the heat transfer through this layer interact with the heat transfer through the railroad grade/road bed during different times of the year? How would lateral heat transfer play a role? Could the different geometries of the roadbed/the railroad grade play a role? Is it certain that winter snow cover can entirely be neglected in the discussion (the authors state that there is no steady or winter-long snow cover)? Is there snow accumulation at the shoulders of the road/railroad? What causes the significant offset between MAAT and MAGT (Table 2) if it is not snow? Is this related to radiative heating of the surface and thereby caused difference between MAAT and MAGST?

-Please revise the English language!

Minor points:

L. 32: How do you define alpine meadow and alpine steppe?

L. 38: Please explain what is meant with "increasing permafrost table"?

[Figure]

L. 54: How are the conclusions of the study influenced by the fact that the boreholes in alpine meadow are at the centerline of the QTR, but at the shoulder for alpine steppe?

Fig. 1: What is meant by "Country" in the figure legends?

l. 107: I don't understand this sentence. What is meant by "cool energy" and why should this be the case?

l. 124: What is meant by "contributed"? Wouldn't it rather be "caused by"?

l. 126: What is meant exactly by "engineering activity increase of APT"?

l. 175: But this vegetation layer will decay and compress over time, thus changing its thermal properties? Is there any evidence how the "vegetation layer" under the railway looks today and how fast this process of decay/compression has occurred/is occurring?

l. 183: I don't think that QTH and QTR are really comparable – one is a road, the other one a railroad grade, with completely different thermal properties. It is thus not necessarily clear that the described effect on ground temperatures is due to the vegetation layer.

l. 217: Why is it a disadvantage? I don't think this follows from this study; at least no evidence for this is presented.

---

## Author Comment (AC2) · 17 Mar 2016

Reply comments on reviewer 2#:

The manuscript" Thermal impacts of engineering activities on permafrost in different alpine ecosystems in Qinghai-Tibet Plateau, China" presents measurements of the ground thermal regime in a range of settings impacted by construction activities. Although the manuscript contains a number of interesting data sets and findings, I do not recommend the manuscript for publication in TC, unless serious revisions are conducted by the authors. In particular, the authors present a number of interpretations for their measurements which are not well enough supported. This could be done by a) a statistical analysis of a sufficient number of samples (= boreholes in this case which is most likely difficult to achieve), or b) a careful, at least semi-quantitative argumentation involving the physics of the system, in particular the heat fluxes related to the thermal properties of the different parts/layers of the system and possibly the radiative forcing at the surface, or c) a study with a 2-dimensional ground thermal model considering the points raised under b).

Reply: We are very thankful for the comments of our manuscript and good suggestion. We try to add data measurement of three boreholes to further support our findings.

Major points:
The manuscript presents and discusses a lot of measurements in great detail (which is OK). However, some of the conclusions on the underlying processes, although not implausible, are not directly supported by the measurements. Although a number of boreholes exist, their number is too small for a statistical analysis that could secure the interpretations given by the authors. On the other hand, no data on the processes themselves are presented. In the following, I comment on the different results and conclusions as given in the abstract (l. 12 ff): "The results show that alpine meadows on the Qinghai–Tibet Plateau can have a controlling role within engineering construction effects on permafrost beneath embankments. "Why not also alpine steppes? Their controlling role would be different, but they would still have a controlling role?

Reply: When railway embankment constructed, the vegetation layer can be remained. The vegetation in alpine meadow can prevent heat conducting, because the vegetation layer in an alpine meadow in Qinghai-Tibet Plateau has thicker humus soils with a small thermal conductivity. Although a vegetation layer of alpine steppe were remained, there is not any humus soil with sparse vegetation. So, soil with sparse vegetation layer has pressed, heat insulation can be found. We adds some interpretation in discussion as follows:

This is because the vegetation layer in an alpine meadow has thicker humus soils with a small thermal conductivity, reducing heat amount conduct down.

"The artificial permafrost table (APT) beneath embankments is predominantly controlled by alpine ecosystems, : : :" Is this not a direct consequence of different ALTs before construction?

Reply: Thank you. It is my fault. I have not clearly show my mean. We revised as following:

As before railway constructed, the artificial permafrost table (APT) beneath embankments is predominantly affected by alpine ecosystems, but the change rate of APT is not closely related with those ecosystems, but dominantly affected by climate change and engineering activities.

": : : but the change rate of APT is not closely related with those ecosystems. " Is it possible to draw this conclusion from the few boreholes, considering that the spatial variability could be quite high?

Reply: Thank you for your comments. We add some data measurement of four boreholes to explain this view. These data newly adding show the same result, seeing in Table 3.

"it is mainly related with cooling effects of railway ballast and heat absorption effects of asphalt pavement." No evidence for this is presented, although it is a plausible conclusion.

Reply: Thank you for your comments. Except the effect of climate change, APT change is contributed to engineering activities. ALT in Qinghai-Tibet Plateau show a continuously increased trend, this result can be documented in many literature (Wu et al., 2012, The Cryosphere; Li, et al., 2012, Chinese Sciences Bulletin; Wu et al., 2015, Global and Planetary Change, listing in the reference). So, the trend of APT decreasing beneath railway embankment and increasing beneath highway can certainly contributed to the cooling effect of railway ballast and heat absorption heat of asphalt pavement because these data from general embankment without any measures of keeping cooling. So, this conclusion is plausible.

"Variation of soil temperature beneath embankments is independent of alpine ecosystems, but variation of mean annual soil temperature with depth is closely related to those ecosystems." It is not clear to me what the authors mean with that and how this could be explained in terms of changing energy content of the soil.

Reply: Thank you for your comments. We may explain unclearly. Variation of soil temperature beneath embankment seems to be difficult to show the difference between alpine meadow and steppe, but we can easily see the difference of mean annual soil temperature with depth. So, we revised it as follows:

Variation of soil temperature beneath embankments is difficult to identify the difference between alpine meadow and alpine steppe, but variation of mean annual soil temperature with depth can be easily found out the difference between alpine meadow and steppe.

"The vegetation layer in alpine meadows can have an insulation role within engineering activity effects on permafrost beneath embankments. " The problem is that the data set does not allow differentiating between the effects of road pavement/railroad grade vs. vegetation removal/no vegetation removal. It is at least possible that the described effects are entirely due to the different heat transfer processes in the roadbed and railroad grade.

Reply: Thank for your comments. We infer that the vegetation layer in alpine meadow can play an insulation role on underlying permafrost based on the mean annual soil temperature with depth. The data set cannot be differentiating this effect of road/railroad vs. vegetation, we need special research design to study this problems. On the different heat transfer processes in the roadbed and railroad grade, we simply explain that ballast pavement of railway has a strong air convection effect, it may have a cooling effects by many literatures, and asphalt pavement has a strong heat absorbed effect by many literatures. We specially study the physical mechanics of asphalt pavement by energy balance, but ballast pavement not. Except pavement has a difference with different heat transfer, the heat transfer of roadbed and railroad grade is same because filled soil is same.

"This insulation role is an advantage for alleviating permafrost temperature rise in the short term," I agree with this finding, the around ten-year time series supports this. "but a disadvantage in the long term because of climate warming, suggesting that vegetation layer in alpine meadow should be removed upon initiating engineering construction" No evidence for this conclusion is presented.

Reply: Thank you for your comments. We cancel this conclusion.

-What happens to the insulating vegetation layer when it is buried under the railroad grade? How thick is this vegetation layer, and how would the heat transfer through this layer interact with the heat transfer through the railroad grade/road bed during different times of the year? How would lateral heat transfer play a role? Could the different geometries of the roadbed/the railroad grade play a role? Is it certain that winter snow cover can entirely be neglected in the discussion (the authors state that there is no steady or winter-long snow cover)? Is there snow accumulation at the shoulders of the road/railroad? What causes the significant offset between MAAT and MAGT (Table 2) if it is not snow? Is this related to radiative heating of the surface and thereby caused difference between MAAT and MAGST?

Reply: Thank you for your comments. You propose many problems being worth to study. Indeed, we cannot answer your problems now, because we may require special study design to study these problems. On the different geometries of the roadbed/the railroad grade, it have strong different heat effect of sunny-shadow slope, there are many paper to study this problems. In Qinghai-Tibet Plateau, no stable snow cover will accumulate at the shoulder of the road/railroad. MAGT is the temperature in the depth of 12 to 15 m beneath ground surface, heat transfer cause the offset between MAAT and MAGT. The offset between MAAT and AMGST is significant as the radiative heating of the surface.

Please revise the English language!

Reply: English language of our manuscript was revised by scientist of native English.

Minor point:
L. 32: How do you define alpine meadow and alpine steppe?

Reply: we define alpine meadow and steppe based on dominant species and vegetation cover.

L. 38: Please explain what is meant with "increasing permafrost table"?

Reply: L. 38, increasing of permafrost table means that permafrost table is deepening.

L. 54: How are the conclusions of the study influenced by the fact that the boreholes in alpine meadow are at the centerline of the QTR, but at the shoulder for alpine steppe?

Reply: Thank you for your comments. Because soil temperature at the centerline of the QTR has not been set from railway pavement to 20m for alpine steppe, we substitute soil temperature at the centerline of the QTR by using data from two shoulder of the QTR. This may have some heat effect of embankment slope, but it cannot change our understanding.

Fig. 1: What is meant by "Country" in the figure legends?

Reply: Sorry, it should be "country".

l. 107: I don't understand this sentence. What is meant by "cool energy" and why should this be the case?

Reply: Thank you for your comments. "cool energy" means the amount of heat release in winter. So, we revised:

"cool energy" is revised into the amount of heat release.

l. 124: What is meant by "contributed"? Wouldn't it rather be "caused by"?

Reply: Thank you. We revised "contributed" into "attributed to"

l. 126: What is meant exactly by "engineering activity increase of APT"?

Reply: Sorry, this sentence seems to be repeated. We cancel it.

l. 175: But this vegetation layer will decay and compress over time, thus changing its thermal properties? Is there any evidence how the "vegetation layer" under the railway looks today and how fast this process of decay/compression has occurred/is occurring?

Reply: Thank you for your comments. I absolutely agree with your opinion. From the view of Fig. 8, the temperature gradient from vegetation layer to a depth beneath embankment is gradually decreasing and trend of permafrost warming is gradually weakening, indicating the heat insulation effect of vegetation will decay. We add some explanation. At the same times, we revised the conclusions by adding the mentioned explanation. Thank you, you give us a right conclusion.

From the view of Fig. 8, the temperature gradient from vegetation layer to a depth beneath embankment is gradually decreasing and trend of permafrost warming is gradually weakening, indicating the heat insulation effect of vegetation will decay.

This suggests that vegetation layer of alpine meadow has an insulation role within the effects of engineering activities on permafrost beneath embankment, but insulation role is gradually disappeared because this vegetation layer will decay and compress over time. On the whole, this vegetation layer is an advantage for alleviating permafrost temperature rise in the short term, but this role is gradually weakened in the long-term.

l. 183: I don't think that QTH and QTR are really comparable – one is a road, the other one a railroad grade, with completely different thermal properties. It is thus not necessarily clear that the described effect on ground temperatures is due to the vegetation layer.

Reply: Thank you for your comments. Although embankment pavement is different, filled soil of embankment is same,

l. 217: Why is it a disadvantage? I don't think this follows from this study; at least no evidence for this is presented.

Reply: Thank you. According to your comments, we re-revise our conclusion. On the whole, vegetation layer is advantage.

---

## Editor Decision (ED1)

[revised manuscript text omitted]
  value (about −0.77 °C at BLR1 in  and −0.36 °C at BLR2 in  However, there was no such finding ( and d) for the QTH without a vegetation layer. Under the combined effect of climate change and engineering activities, soil temperature in the upper vegetation layer  trend,  temperature in a certain depth range    beneath the vegetation layer

had an  increasing trend for the railway with a vegetation layer in alpine meadow (Figure 8a and b and Table 4).  soil temperature at all depths showed  increasing trends for the highway with vegetation layer  (Figure 8c and d and Table 4). These  indicate that the vegetation layer in an alpine meadow may have an insulating role among the effects of engineering activities on permafrost beneath embankments.  analyzed changes of permafrost thermal regime beneath embankments in alpine steppe, for both QTR with a vegetation layer and QTH without that layer.  no pattern similar to that in Figure 8a and b, demonstrating that that vegetation layer has no insulation effect.  the vegetation layer in an alpine meadow can effectively prevent heat disturbance from engineering construction from propagating rapidly downward and raising permafrost temperature over the short term. This is because  humus soils with small thermal conductivity, reducing downward heat .

[Figure]

**Figure 8 Change of mean annual soil temperature with depth beneath embankment at BLR1 (a) and BLR2 (b), BLR3 (c), and BLR4 (d) for QTR, and at TGH1 (e), FHH1 (f), and KXH1 (g) for QTH in alpine meadow**

Generally, an insulation layer within an embankment can mitigate heat disturbance from short-term engineering activities (Esch, 1987; Cheng et al., 2004; Wen et al., 2005). However, such a layer is a disadvantage  long-term effects of climate warming over the period of engineering operation (Liu et al., 2002; Sheng et al., 2006), especially for warm permafrost

Although we cannot know what happens to an insulating vegetation layer after it is buried under a railroad grade, we can infer that this layer is compressed over time, altering its thermal properties in an alpine meadow. As a consequence, its insulation effect may gradually weaken (McHattie and Esch, 1983). From 💬 8, the temperature gradient from the vegetation layer to a given depth beneath the embankment gradually decreases, and the trend of permafrost warming gradually weakens between 3

5  and 4 years after railway construction. This indicates that the heat insulation effect of vegetation changes.

Generally, the vegetation layer in alpine meadow of the Qinghai-Tibet Plateau, including the humus and root-layer soils, is thin, with maximum thickness < 60 cm (Li et al., 2007). Vegetation roots mainly reach depths of ~10 cm, and mean root biomass makes up 60% of total root biomass (Yue et al., 2015). After the railway or highway embankment is constructed, soil within the vegetation layer of the alpine meadow is compressed and soil moisture decreases, modifying soil heat transfer.

10  Because heat conductivity within the vegetation layer of alpine meadow from humus soil is less than that of filled soil of embankments, the vegetation layer can effectively prevent downward heat transfer, decreasing the amount of heat in permafrost. Meanwhile, moisture within the alpine meadow vegetation layer migrates upward in the embankment soil. At present, we cannot quantitatively analyze such a process,  heat and moisture transport through the vegetation may be affected by lateral heat transfer, different geometries of the

15  roadbed/railroad, and snow  on the lateral embankment slopes.

The effect of lateral heat transfer on permafrost beneath embankments has two sources.  horizontal heat exchange  embankment, and  The horizontal heat exchange is generally small  to soil heat conduction. However,  strongly influences permafrost beneath  
[revised manuscript text omitted]

---

## Author Response (AR2)

**Editor Decision: Reconsider after major revisions** (11 May 2016) by Prof Christian Hauck Comments to the Author: Dear authors,

many thanks for your revised version and answers to the reports of the two reviewers. Due to the nature of the major comments of one of the reviewers, the manuscript was again sent out for review. This second review highlighted once more the two main concerns that still remain, also from my point of view:

(1) There are still (and even more so after your revision) some serious problems with the english language. Whereas the main part of the original manuscript can be well understood, this is not the case for most of your revision and also your answers to the reviewers comments. Sometimes it is not possible to understand at all what you want to say, e.g. the new lines in the abstract; also the lines 1-3 on page 6; and as well the new text on pages 11 and 12. Without an improvement of the english of the whole manuscript, at least to a point where the meaning of all sentences can be understood, the manuscript cannot be published. After this improvement, we can provide help for the final polishing of the english.

Reply: Thank you. I edited my revised manuscript by specialist of native English. Please seeing the attached file.

(2) In the original review the reviewer has formulated several questions that should be addressed in the manuscript: "What happens to the insulating vegetation layer when it is buried under the railroad grade? How thick is this vegetation layer, and how would the heat transfer through this layer interact with the heat transfer through the railroad grade/road bed during different times of the year? How would lateral heat transfer play a role? Could the different geometries of the roadbed/the railroad grade play a role? Is it certain that winter snow cover can entirely be neglected in the discussion (the authors state that there is no steady or winter-long snow cover)? Is there snow accumulation at the shoulders of the road/railroad? What causes the significant offset between MAAT and MAGT (Table 2) if it is not snow? Is this related to radiative heating of the surface and thereby caused difference between MAAT and MAGST?"

Although it is clear that most of these questions cannot be definitely answered within the present study (as you pointed out in your answer), their relative importance and/or their potential impact on the uncertainty of the results of your study can be discussed. These uncertainties should at least be mentioned, and their effect may be estimated using results from other studies. The reviewer suggested to include a discussion on the possible magnitude of each effect, e.g. how much additional heating and thus difference between MAAT and MAGST can one expect from radiative heating on an asphalt surface? Are there numbers for this effect on the Qinghai-Tibet Plateau presented in other publications? Furthermore, the author's response on the effect of highway/railroad geometries could be included in the manuscript, including references to the relevant studies.

Reply: Thank you for your comments and good suggestions. These problems is very important by editor and reviewer propose. We do our best to discuss these problems in discussion. The following text are added in discussion.

Generally, the vegetation layer in alpine meadow of the Qinghai-Tibet Plateau, including the humus and root-layer soils, is thin, with maximum thickness < 60 cm (Li et al., 2007). Vegetation roots mainly reach depths of ~10 cm, and mean root biomass makes up 60% of total root biomass (Yue et al., 2015). After the railway or highway embankment is constructed, soil within the vegetation layer of the alpine meadow is compressed and soil moisture decreases, modifying soil heat transfer. Because heat conductivity within the vegetation layer of alpine meadow from humus soil is less than that of filled soil of embankments, the vegetation layer can effectively prevent downward heat transfer, decreasing the amount of heat in permafrost. Meanwhile, moisture within the alpine meadow vegetation layer migrates upward in the embankment soil, redistributing its moisture. At present, we cannot quantitatively analyze such a process, so focused study is needed in that area. This process of heat and moisture transport through the vegetation may be affected by lateral heat transfer, different geometries of the roadbed/railroad, and snow cover on the lateral embankment slopes.

The effect of lateral heat transfer on permafrost beneath embankments has two sources. One is from horizontal heat exchange outside the embankment, and the other is a heat effect of its slope. The horizontal heat exchange is generally small owing to soil heat conduction. However, lateral convection heat transfer strongly influences permafrost beneath the embankment. Water flow can especially accelerate permafrost thaw (Grandpr éet al., 2012). The heat effect of embankment slope on permafrost beneath the embankment is mainly from the thermal effect of sunny-shade slope (Chou et al., 2008a). The resulting difference in solar radiation has a thermal effect on the sunny and shaded slopes of embankments constructed within permafrost regions, producing differences in soil temperature and the permafrost table under the shoulder (Chou et al., 2008b; Wu et al., 2011). Monitoring data of soil temperature along the QTR show that the difference in temperature and APT between sunny and shaded slopes of the embankment at WD3, KL1 and KL3 in alpine meadow (Wu et al., 2012) is generally small, < 1 °C and 20 cm, respectively (Wu et al., 2011), but that difference in alpine steppe is > 1.5-3.0 °C and 100-300 cm (Wu et al., 2011). These results may indicate that the alpine meadow vegetation layer beneath embankments reduces differences in soil temperature and APT under the shoulder. However, a large embankment height strengthens that difference, because of greater radiation on the sunny slope (Hu, 2006). The varying geometries of the roadbed/railroad have a thermal effect on permafrost beneath the embankment. The embankment width affects the annual heat transfer rate at the bottom of the embankment (Yu et al., 2007). The annual rate increased by 60% with doubling of the width of asphalt pavement (Yu et al., 2007). This increased rate was mainly at the bottom of the embankment, resulting in thermal concentration. Therefore, substantial heat enters the permafrost through the vegetation layer.

On the Qinghai-Tibet Plateau, snow mainly accumulates in the high mountains, with little in the plateau interior (Li and Mi, 1983; Sun et al., 2014). Snow cover is generally thin, less than 6 cm on average, and the duration of cover is short (Li and Mi, 1983; French, 2007; Tian et al., 2014). The insulation of snow cover is weak when it is

**Certificate of English Editing**

| Date of Issue         | 13 June 2016                                                                                                                                                                |
|-----------------------|-----------------------------------------------------------------------------------------------------------------------------------------------------------------------------|
|                       |                                                                                                                                                                             |
| About the manuscript: |                                                                                                                                                                             |
| Title                 | Thermal impacts of engineering activities and vegetation layer on permafrost in different alpine ecosystems of Qinghai-Tibet Plateau, China                                 |
| First Author          | Wu Qingbai                                                                                                                                                                  |
| Affiliation           | State Key Laboratory of Frozen Soil Engineering, Cold and Arid Regions Environmental and Engineering Research Institute, Chinese Academy of Science, Lanzhou, 730000, China |
| Date of editing       | 13 June 2016                                                                                                                                                                |

| About the editor:
Editor | Steven Hunter
1983 - M.S. Atmospheric Science - University of Wyoming
Expert in all fields relating to meteorology |  |
|------------------------------------|--------------------------------------------------------------------------------------------------------------------------|--|
|                                    | Full profile                                                                                                      |  |

Certificate issued by

Benjamin Shaw Director

Liwen Bianji (Edanz Group China)

blijamin Shew

While this certificate confirms the authors have used Edanz's editing services, we cannot guarantee that additional changes have not been made after our edits.

---

## Author Response (AR3)

**Reply to Reviewers' comments on manuscript submitted to The Cryosphere by Wu et al.,**

First of all, we appreciate Professor Christian Hauck for his constructive and insightful comments and suggestions for this manuscript. We consider all comments and suggestions seriously. All comments are very helpful for further revision of this manuscript. We have made all changes based on the reviewers' comments and suggestion as described below.

We use the red to identify our revised text in our manuscript. Please seeing our manuscript revised.